# Comparative Transcriptome Analysis Reveals Key Insights into Fertility Conversion in the Thermo-Sensitive Cytoplasmic Male Sterile Wheat

**DOI:** 10.3390/ijms232214354

**Published:** 2022-11-18

**Authors:** Zihan Liu, Fuqiang Niu, Shaohua Yuan, Shuying Feng, Yanmei Li, Fengkun Lu, Tianbao Zhang, Jianfang Bai, Changping Zhao, Liping Zhang

**Affiliations:** 1Beijing Key Laboratory of Molecular Genetics in Hybrid Wheat, Institute of Hybrid Wheat Beijing Academy of Agriculture and Forestry Science, Beijing 100097, China; 2College of Agronomy, Northwest A&F University, Xianyang 712100, China; 3Blue Red Hybrid Wheat Research Center, Xianyang 044000, China

**Keywords:** fertility conversion, cytoplasmic male sterility, transcriptome sequencing, hybrid wheat

## Abstract

Thermo-sensitive cytoplasmic male sterility (TCMS) plays a crucial role in hybrid production and hybrid breeding; however, there are few studies on molecular mechanisms related to anther abortion in the wheat TCMS line. In this study, FA99, a new wheat thermo-sensitive cytoplasmic male sterility line, was investigated. Fertility conversion analysis showed that FA99 was mainly controlled by temperature, and the temperature-sensitive stage was pollen mother cell formation to a uninucleate stage. Further phenotypic identification and paraffin section showed that FA99 was characterized by indehiscent anthers and aborted pollen in a sterile environment and tapetum was degraded prematurely during the tetrad period, which was the critical abortion period of FA99. The contents of O^2−^, H_2_O_2_, MDA and POD were significantly changed in FA99 under a sterile environment by the determination of physiological indexes. Furthermore, through transcriptome analysis, 252 differentially expressed genes were identified, including 218 downregulated and 34 upregulated genes. Based on KOG function classification, GO enrichment and KEGG pathways analysis, it was evident that significant transcriptomic changes in FA99 under different fertility environments, and the major differences were “phenylalanine metabolism”, “phenylpropanoid biosynthesis”, “cutin, suberine and wax biosynthesis”, “phenylalanine, tyrosine and tryptophan biosynthesis” and “citrate cycle (TCA cycle)”. Finally, we proposed an intriguing transcriptome-mediated pollen abortion and male sterility network for FA99. These findings provided data on the molecular mechanism of fertility conversion in thermo-sensitive cytoplasmic male sterility wheat.

## 1. Introduction

Wheat (*Triticum aestivum* L.) is one of the most widely grown food crops in the world, providing 21% of food calories and 20% of protein sources for about 35% of the world’s population. With the growth of population and urbanization and the continuous reduction in the planting area, the gap between wheat production and demand is growing, and the prospect of wheat production is worrisome [1]. Studies have shown that the utilization of wheat heterosis is an effective way to obtain excellent traits such as high yield, high resistance, high quality and strong adaptability [2]. Male sterility is the most important way to utilize heterosis in wheat [3]. According to inheritance patterns of male sterility, it can be divided into cytoplasmic male sterility (CMS), genic male sterility (GMS) and photo-thermo-sensitive cytoplasmic/genic male sterility (PTC/GMS) [4]. The two-line hybrid wheat technology system, which utilizes the (PTC/GMS) line of wheat to produce hybrids, has become the mainstream technology for the utilization of wheat heterosis. However, the research on the genetic mechanism of (PTC/GMS) is still relatively lagging behind in wheat.

Male sterility is mainly caused by abnormal degradation of the tapetum, pollen wall defects, and anther dehiscence [3]. The tapetum, located in the innermost anther wall, provides sufficient nutrients and secretes corpus callosum enzymes, recognition proteins and sporopollen for pollen development; however, abnormal activity of the tapetum can lead to male sterility [5,6]. In *Arabidopsis*, *MS1* encodes the PHD transcription factor. Its mutant *ms1* displays tapetum vacuolation and delayed degradation of PCD, resulting in male sterility [7,8]. *DYT1*, a bHLH family transcription factor, is specifically expressed in tapetum cells. In *dyt1* mutants, tapetum cells show severe vacuolation and non-specific thickening during meiosis, leading to pollen abortion [9]. In addition, *AMS* and *TDF1* are also remarkably important in tapetum cells PCD [10,11]. On the other hand, normal pollen wall formation is a prerequisite for male gamete development. In rice, the mutation of *EDT1*, which encodes an ATP-citrate lyase, exhibits defective pollen walls and complete male sterility [12]. *OSGT1,* encoding a glycosyltransferase, is essential for pollen wall formation, and this mutant displays defective intine formation [13]. In wheat, *TaPG* encodes a polygalacturonase and significantly abounds in fertile anthers. The silencing of *TaPG* displays exhibits indehisced anthers and defective pollen wall formation [14]. Notably, more and more studies have identified intracellular factors that regulate the transcriptional network of tapetum PCD, and it is believed that reactive oxygen species (ROS) act as important signaling molecules in the tapetum PCD [15,16]. Mitochondria play a crucial role in maintaining the normal function of cells, including the response to oxidative stress signals, the initiation of PCD and the synthesis of nucleic acids and proteins [17]. Therefore, the accumulation of ROS and the abnormal development of tapetum and pollen wall are closely related to male sterility, but their molecular mechanism in male sterility wheat is still unclear.

At present, transcriptome sequencing is an indispensable part of systems biology. Transcriptome sequencing can reflect the expression in cells and tissues at the overall level, and is widely used in many fields [18]. Male sterility is a common biological phenomenon, and a large number of genes are involved in the regulation. Transcriptome analysis can explore key genes and metabolic pathways and provide new ideas for the mechanism of male sterility. In eggplants, 1716 differentially expressed genes (DEGs) are associated with pollen development [19]. In cabbages, many key DEGs are involved in pollen wall formation and tapetum development [20]. In wheat, 3420 DEGs related to male sterility are mainly involved in phenylpropanoid biosynthesis pathways [3]. The phenylpropanoid biosynthesis pathway plays a pivotal role in the development of pollen. The key enzymes, phenylalanine ammonia-lyase and cinnamyl alcohol dehydrogenase, are involved in the formation of sporopollenin and provide sufficient nutrients for the development of pollen walls [21]. Studies have shown that the normal expression and regulation of genes are the basis for pollen formation and development in the phenylpropane biosynthetic pathway. When the gene expression changes, this pathway is disrupted, resulting in male sterility [22].

In this study, the new TCMS line FA99 was used as material to investigate its fertility conversion, phenotypic characteristics, cytological observation and physicochemical properties. Furthermore, RNA-Seq was conducted using the anthers from FA99 and J411 under different fertility conditions to identify DEGs and the key metabolic pathways. This study is expected to provide a theoretical basis and novel insights into the fertility conversion mechanism in TCMS wheat lines.

## 2. Results

### 2.1. Fertility Conversion Analysis of FA99

In the phased sowing experiment at different ecological points, we found the seed setting rate of FA99 in Shunyi, Beijing was significantly higher than that in Dengzhou, Henan. Moreover, the seed-setting rates of FA99 sowing late were all higher than those of early sowing in Shunyi and Dengzhou. In Dengzhou, Henan Province, the seed setting rates of FA99 sowing on 30 September and 10 October were significantly different from that of FA99 sowing on 20 October, 30 October and 10 November. The seed setting rate of FA99 sowing from 30 September to 10 October is less than 1%, reaching high sterility, which is a safe seed production and sowing period. In Shunyi, Beijing, the seed setting rate of FA99 sowing on November 10 was the highest and significantly higher than that before 10 October, indicating that the seed setting rate of FA99 sowing on 10 November can be safely reproduced. Although the seed setting rate of control variety J411 late sowing was slightly higher than that of early sowing in different regions, which reached more than 90%, the difference was not significant. The above analysis showed that the environment has little effect on the fertility of J411, and the fertility performance is normal, whereas the fertility of FA99 was significantly affected by different ecological areas and sowing dates and it obviously exhibited a fertility conversion trait (Table 1).

To further identify the photo-thermo-sensitive effect of FA99 fertility conversion, photo-thermal regulation assays were performed. In the temperature-sensitive period test, the control J411 had no significant difference in seed-setting rate among different temperature treatments, but the temperature had a significant effect on FA99. The seed-setting rate of FA99 at the average temperature of 10 °C and 12 °C was lower than 1%, and the difference between the two treatments was not significant. While the seed setting rates of the treatments with an average temperature of 14 °C and 16 °C reached 8.51% and 19.36%, respectively, which were significantly higher than the average temperature of 10 °C and 12 °C. Therefore, the temperature threshold of FA99 was 12–14 °C under the condition of 12 h·d^−1^ light. However, in the light-sensitive period test, the TCMS of FA99 is insensitive to light in cold conditions. Therefore, we demonstrated that temperature is the main factor that affected FA99 fertility and indicated FA99 is a low temperature-induced male sterility line. In addition, in order to explore the temperature-sensitive period of FA99 fertility conversion, the anthers at different developmental stages were treated in a low temperature (12 °C). The results showed that the seed-setting rate from the pollen mother cell stage to the uninucleate stage was significantly lower than floret differentiation to pollen mother cell formation, which indicated that the temperature-sensitive period of FA99 was the pollen mother cell stage to the uninucleate stage (Table 2).

### 2.2. Phenotypic Characteristics for FA99 at Different Environments

In order to observe the fertility differences of FA99 in different environments, the stamens and pistils of FA99 and J411 at the trinucleate stage were taken for stereoscopic observation. There was no difference in anthers and pistils of J411 at the trinucleate stage under different fertility environments. It was found that the pistils of FA99 and J411 had no morphological difference. However, compared with B-FA99 (FA99 in the fertile condition), B-J411 (J411 in the fertile condition) and D-J411 (J411 in the sterile condition), the anthers of D-FA99 (FA99 in the sterile condition) did not dehiscence and no pollen was released (Figure 1A–D). I_2_-KI staining showed that the pollen grains of B-FA99, B-J411 and D-J411 were dyed black, while the pollen grains of D-FA99 showed shrinkage and deformity, and the most of pollen grains were unstained (Figure 1E–H). The above results showed that environmental differences greatly affected the development of FA99 anthers and microspores.

To further clarify the key stage and cytological causes of FA99 fertility conversion, paraffin sections were applied to observe the dynamic process of anthers development during the six different developmental stages (microspore mother cell stage, dyad stage, tetrad stage, uninucleate stage, binucleate stage and trinucleate stage). At the microspore mother cell stage (MMCs) and dyad stage (DYAd), there were no significant morphological differences in the anther locules, which are composed of the tapetum, middle layer, endothecium, and epidermis from inside to outside. At the tetrad stage (Td), compared with B-FA99, B-J411 and D-J411, the tapetum cells of D-FA99 began to degrade significantly while tapetum cells of B-FA99, B-J411 and D-J411 had a wide outline. At the uninucleate stage (Un), the tapetum cells of D-FA99 continued to degenerate, and the microspore vacuolized and shrinkage. At the binucleate stage (Bn), the tapetum layer of D-FA99 was almost completely degraded, whilst the B-FA99, B-J411 and D-J411 had a visible tapetum layer. Until the trinucleate stage (Tn), the anther wall was dehisced to release mature pollen grains in B-FA99, B-J411 and D-J411. In contrast, the anther locule remained closed with no pollen release in the FA99 (Figure 2). Taken together, the results showed that the tetrad stage could be the key fertility conversion stage for FA99, and the early degradation of tapetum may be an important reason for pollen abortion and male sterility.

### 2.3. Determination of Physiological Indexes

More and more evidence shows that the abnormal degradation of tapetum may be related to reactive oxygen species (ROS) [15,23]. In order to further explore the potential causes of abnormal tapetum degradation, the O^2−^, H_2_O_2_, malondialdehyde (MDA) and peroxidase (POD) contents were determined in anthers at different stages in B-FA99 and D-FA99. The results showed that the contents of O^2−^, H_2_O_2_ in D-FA99 were higher than those in B-FA99, and reached the peak values at the tetrad stage (Figure 3A,B). The excessive accumulation of ROS leads to the peroxidation of unsaturated fatty acids in the anther cell membrane, which is degraded into small molecular substances MDA. Starting with the tetrad stage, the MDA content of D-FA99 was significantly higher than that of B-FA99 (Figure 3C). In addition, POD plays an important role in the metabolism of cellular ROS. From the tetrad stage, the POD activity of D-FA99 is significantly lower than that of B-FA99. POD activity was obviously down-regulated at the tetrad stage, thereby resulting in a decrease in the ability of scavenging ROS (Figure 3D). The above results indicated that excessive ROS disrupted the balance of the antioxidant system and the presence of excess ROS may have been related to premature degradation of the tapetum, which eventually led to the abortion of FA99.

### 2.4. Transcriptome Analysis

To explore the molecular mechanisms responsible for the abnormal accumulation of ROS, premature degradation of the tapetum and pollen abortion in the FA99, three individual biological replicates of anthers in the tetrad stage from FA99 and J411 under different fertility environments were used for transcriptome sequencing analysis. In total, 12 libraries (D-FA99-TE-1, D-FA99-TE-2, D-FA99-TE-3, B-FA99-TE-1, B-FA99-TE-2, B-FA99-TE-3, D-J411-TE-1, D-J411-TE-2, D-J411-TE-3, B-J411-TE-1, B-J411-TE-2, B-J411-TE-3) were sequenced and 724,965,558 clean reads were obtained, with 501,178,932 reads from FA99 and 223,786,626 from J411. The percentage of Q30 exceeded 92.84%, and the GC content for the clean data ranged from 53.33% to 56.94%. The alignment efficiency of clean reads from each sample ranged from 84.094% to 94.16% compared with the wheat reference genome (Table 3). The results showed that the quality of the transcriptome sequencing was reliable and sufficient for further analysis.

### 2.5. Screening of Differentially Expressed Genes

Differentially expressed genes (DEGs) were detected based on screening criteria (|log_2_(foldchange)| ≥ 1, *p*-value < 0.05) in FA99 and J411 (D-FA99 vs. D-J411, B-FA99 vs. B-J411, B-J411 vs. D-J411, B-FA99 vs. D-FA99). Overall, 17,430 genes were obtained in FA99 and J411 under different fertility environments. After removing background noise (D-FA99 vs. D-J411, B-FA99 vs. B-J411, B-J411 vs. D-J411), 252 DEGs were obtained in B-FA99 vs. D-FA99, comprising 218 downregulated DEGs and 34 upregulated DEGs (Figure 4, Appendix A). We found that most of the DEGs were downregulated which may be related to male sterility in D-FA99.

### 2.6. KOG and Gene Ontology (GO) Analysis of DEGs

To further clarify the functional genes, KOG classifications waweres employed. In total, 93 successfully annotated DEGs can be divided into three categories according to the 25 functional categories of KOG. The results showed that the DEGs were involved in processes including “transcription”, “posttranslational modification, protein turnover, chaperones” and “carbohydrate transport and metabolism”. These processes may play important roles in the anther development and differentiation in FA99 (Appendix A). According to GO function classification analysis, the DEGs annotated were classified into 33 functional groups in three principal categories comprising “biological process”, “cellular component” and “molecular function”. In the biological process, the DEGs were primarily associated with the metabolic process (GO: 0008152), cellular process (GO: 0009987), biological regulation (GO: 0065007) and regulation of biological process (GO: 0050789). The main cellular component contained 10 GO terms. Most of the DEGs were primarily concerned with cell (GO: 0005623), cell part (GO: 0044464), membrane (GO: 0016020), organelle (GO: 0043226) and membrane part (GO: 0044425). Binding (GO: 0005488) and catalytic activity (GO: 0003824) were dominant in the category of molecular function (Figure 5A). In addition, the top 20 enriched GO terms were rich in the “biological process” term (Figure 5B). We also found there were more downregulated DEGs than upregulated DEGs in FA99 based on the Z-score bubble diagram, and the downregulated DEGs significantly enriched in the regulation of transcription, DNA-templated (GO:0006355), site of polarized growth (GO:0030427), DNA–binding transcription factor activity (GO:0003700), respectively from “biological process”, “cellular component” and “molecular function” (Figure 5B). The results showed that the DEGs annotated in GO enrichment analysis may provide valuable insight for studying male sterility in FA99.

### 2.7. Kyoto Encyclopedia of Genes and Genomes (KEGG) Pathways of DEGs

In order to identify the functions of DEGs, KEGG pathways enrichment analysis was performed. The DEGs in the FA99 were assigned to 37 pathways in the KEGG database, where five enrichment parts were identified, including “cellular processes”, “environmental information processing”, “genetic information processing”, “metabolism” and “organismal systems”. The most DEGs were associated with “metabolism” comprising “amino acid metabolism”, “biosynthesis of other secondary metabolites”, “lipid metabolism”, “glycan biosynthesis and metabolism”, “metabolism of other amino acids”, “carbohydrate metabolism”, “nucleotide metabolism” and “metabolism of cofactors and vitamins” (Figure 6A). Among the top 20 pathways, many pathways were related to “phenylalanine metabolism”, “phenylpropanoid biosynthesis”, “cutin, suberine and wax biosynthesis”, “phenylalanine, tyrosine and tryptophan biosynthesis” and “citrate cycle (TCA cycle)” based on *p*-value < 0.05 (Figure 6B,C). Previous studies have reported that genes and enzymes related to “phenylalanine metabolism”, “phenylpropanoid biosynthesis” and “phenylalanine, tyrosine and tryptophan biosynthesis” have an important role in male sterility [3,22,24]. Hierarchical clustering analysis was employed to study the nine DEGs in the three pathways. Apart from the genes encoding peroxidase (*TraesCS7D02G212900*) and β-glucosidase (*TraesCS2A02G478500*), seven DEGs including the genes encoding tyrosine aminotransferase (*TraesCS4D02G153400*), phenylalanine ammonia-lyase (*TraesCS2A02G212900*, *TraesCS1A02G094900*), peroxidase (*TraesCS2A02G084100*, *TraesCS1D02G079800*) and peroxygenase (*TraesCS6B02G326500*, *TraesCS6A02G296700*) in the major pathways were downregulated in the D-FA99 compared with B-FA99 (Figure 7A). The results showed these genes may be related to male sterility in FA99.

### 2.8. Confirmation of Key DEGs by qRT-PCR

In order to verify the accuracy of RNA-seq data, the above nine key DEGs were identified using quantitative real-time polymerase chain reaction (qRT-PCR). The results showed that expression patterns of the key DEGs were consistent with the obtained by qRT-PCR (Figure 7B). Overall, the results demonstrated that the RNA-Seq results were accurate and trustworthy.

## 3. Discussion

### 3.1. FA99 Is a Novel Temperature-Sensitive Cytoplasmic Male Sterile Line

At present, the use of wheat male sterile lines to produce hybrids is an effective way to utilize wheat heterosis. The in-depth study of the relationship between environmental induction and fertility of wheat sterile lines is the basis for guiding the selection and application of sterile lines. For the thermo-sensitive sterile lines, the higher the critical temperature, the safer seed production, and the larger the critical threshold span, the wider the selection of ecological regions. In this study, the wheat thermo-sensitive cytoplasmic male sterile line FA99 was used as the research material, and the fertility conversion characteristics of FA99 were analyzed through light and temperature regulation experiments in an artificial climate box. The results showed that FA99 is mainly temperature sensitive, and its temperature conversion threshold is 12–14 °C when exposed to light for 12 h·d^−1^. The fertility conversion mode of FA99 is similar to that of BS210, C49S, A3314, BNS, ES-50 and the temperature conversion threshold of C49S is the same as that of 12–14 °C, while the conversion thresholds of FA99 are different from those of BS210 (8~12 °C), A3314 (18 °C), BNS (8~12 °C) and ES-50 (10.5~13 °C) [25,26,27,28,29,30]. The results provide a theoretical basis for the selection and practical application of FA99 safe seed production ecoregions.

### 3.2. The Critical Period of Abortion in FA99

In angiosperms, the release of normally developing pollen from the anthers is critical for successful insemination. However, many studies have shown that the period of abortion varies among different thermo-sensitive male sterility lines. In the thermo-sensitive sterile line C49S, the period of fertility conversion occurs during meiosis and microsporogenesis [31]. In the thermo-sensitive sterile line ES-4, the fertility conversion stage is the late uninucleate stage, and microspores could not be formed in the subsequent stage [32]. In sterile line BS20, pollen abortion occurs from the pollen mother cell stage to the uninucleate stage [33]. In addition, the fertility conversion of BS366 and BS210 are mainly due to temperature regulation. The meiotic stage and uninucleate stage are sensitive to BS366 and BS210, respectively [25,34]. In this study, the temperature-sensitive characteristics were studied through a temperature-controlled experiment, and the results showed that the fertility conversion period of FA99 was pollen mother cell formation to the uninucleate stage. In order to further study the detailed pollen abortion period of FA99, paraffin sections were used to observe the microstructure of FA99 in seven different developmental stages under different fertility environments. The results showed that in the tetrad stage, the tapetum development of D-FA99 was significantly different from that in B-FA99 and J411, and the tapetum of D-FA99 was degraded prematurely leading anther abortion. In summary, it can be concluded that the critical abortion period for FA99 is the tetrad stage.

### 3.3. Possible Transcriptome-Mediated Male Sterility Network in FA99

According to the KEGG cluster analysis of DEGs, we analyzed the DEGs in the five important metabolic pathways (starch and sucrose metabolism, TCA cycle, phenylalanine, tyrosine and tryptophan biosynthesis, phenylpropanoid biosynthesis and cutin, suberine and wax biosynthesis), as well as considering physiological index determination, cytological observation and previous studies, we propose a possible transcriptome-mediated male sterility network in FA99 (Figure 8). Previous studies have shown that defects in the TCA cycle may disturb the electron transport chain and thereby lead to producing excessive amounts of ROS [35,36]. Moreover, excessive amounts of ROS may induce changes in mitochondrial permeability. It is generally considered that excessive ROS may act as a signal to cause abnormal programmed cell death (PCD) of tapetum cells [37]. In addition, the formation of pollen walls is a complex biological process, and defective cell walls often lead to pollen abortion [38]. The complete pollen wall mainly comprises exine and intine. The exine can be divided into the outer sexine and inner nexine [14]. Sporopollenin, the most important component constituting the exine, comprises a series of highly resistant biopolymers, which are produced by phenylpropanoid and fatty acid biosynthesis pathways [39]. Phenylalanine ammonia-lyase (PAL), peroxidase (POD), and tyrosine aminotransferase (TAT) are the key enzymes in the phenylpropanoid metabolic pathway. Phenylpropane metabolism affects cell wall formation through the lignin metabolic pathway and flavonoid metabolic pathway. In this study, based on the above previous studies, related experimental validation and transcriptome analysis, we constructed a putative network model of male sterility. As shown in Figure 8, low temperature may trigger the upregulation of β-glucosidase (bglX) in the starch and sucrose metabolism, which affects the TCA cycle to some extent. More specifically, upregulation of the DEGs encoding aconitate hydratase (ACO) in the TCA cycle may disturb the electron transport chain to generate excessive amounts of ROS. Meanwhile, the downregulated expression of the DEGs encoding POD and peroxygenase (PXG) and reduced activity of active oxygen-scavenging enzymes mean that ROS cannot be eliminated sufficiently rapidly and then cells undergo oxygen stress-inducing abnormal PCD of the tapetum. Furthermore, the downregulation of the PAL, POD and TAT in the phenylpropanoid biosynthesis pathway may affect the synthesis of lignin, sporopollenin and flavonoid, thereby affecting phenolic polymer synthesis. In the cutin, suberine and wax biosynthesis pathways, the key enzyme (PXG) was downregulated, which resulted in the abnormal synthesis of long-chain fatty acids and reduced lipid polymer synthesis. Phenolic and lipid polymers are important components of sporopollenin. Inhibition of their synthesis and abnormal PCD of tapetum will lead to defects in pollen wall formation and pollen abortion.

## 4. Materials and Methods

### 4.1. Plant Materials

In this study, FA and J411 were used as research materials, which were provided by Shanxi Yuncheng Blue Red Hybrid Wheat Research Center and Institute of Hybrid Wheat, Institute of Hybrid Wheat Beijing Academy of Agriculture and Forestry Science, respectively. The FA is a new F-type CMS line wheat with a common cytoplasm derived from cross and backcross among common wheat varieties. From 2015 to 2017, through further increasing selection pressure and multi-ecological area identification, we screened a thermo-sensitive F-type cytoplasmic male sterile line (named as FA99) whose fertility is greatly affected by temperature, exhibited male sterility in Dengzhou (32.686° N, 112.090° E) and male fertile in Beijing (39.944° N, 116.287° E). During April 2018, FA99 was checked by bagging and the self-setting rate was 1% in Dengzhou and 40% in Beijing, respectively. The anthers from FA99 and J411 in different fertility environments (FA99 and J411 in a sterile environment were denoted as D-FA99 and D-J411, while FA99 and J411 in a fertile environment were denoted as B-FA99 and B-J411) used for RNA-Seq at the tetrad stage (Td) were collected, snap-frozen in liquid nitrogen and stored at −80 °C with three biological replications each sample. Moreover, anthers from D-FA99, B-FA99, D-J411 and B-J411 at the microspore mother cell stage (MMCs), dyad stage (DYAd), tetrad stage (Td), uninucleate stage (Un), binucleate stage (Bn) and trinucleate stage (Tn) were stored in formalin-acetic acid-alcohol (FAA) and glutaraldehyde solution for phenotypic observations.

### 4.2. Photo-Thermo-Sensitive Identification

During the wheat growing season from 2017 to 2019, the FA99 and J411 were sown in Shunyi, Beijing (32.686° N, 112.090° E) and Dengzhou, Henan (39.944° N, 116.287° E) by means of sowing date test, the sowing periods were 30 September, 10 October, 20 October, 30 October and 10 November, respectively. Moreover, the light and temperature regulation experiments were carried out in an artificial climate chamber (KBW, Binder, Germany) from October 2019 to May 2020, with a humidity of 70% and a light intensity of 125 μmol m^−2^ s^−1^. The materials were sown in flower pots on 1 October, and placed in the field. The specifications of the flower pots were 26 cm in diameter at the bottom, 23 cm in diameter at the upper part and 25 cm in height, with 10 seedlings per pot. After the vernalization was completed, the flowerpots were moved into the greenhouse until the seedlings grew to the desired growth period, and then moved into the artificial climate box for different light and temperature treatments. There were three pots per treatment, ten plants per pot and three replicates were set.

Four temperature levels were set in the temperature-sensitive test, the average temperature of the treatment was 10, 12, 14 and 16 °C, and the duration of the treatment was 12 h·d^−1^. Three light length levels were set in the light sensitivity test, the treatment light duration was 10 h·d^−1^, 12 h·d^−1^ and 14 h·d^−1^, and the average temperature of the treatment was 12 °C. The above treatment periods were from pollen mother cell formation to the uninucleate stage. In addition, four treatment periods were set up in the temperature-sensitive period experiment, which were forest differentiation to pistil and stamen differentiation, pistil and stamen differentiation to pollen mother cell stage, pollen mother cell stage to tetrad stage and tetrad stage to uninucleate stage, respectively. The temperature-sensitive period test was performed at a low temperature (average temperature 12 °C), and the light duration was 12 h·d^−1^. The plants were bagged before flowering, with 3 bags per plant, and the seed setting rate was investigated after maturity.

Seed setting rate = number of grains per spike/(number of spikelets × 2) × 100%. The SPSS 20.0 software (SPSS, Chicago, IL, USA) package was used for statistical analysis, and the statistical comparison of multiple sets was performed by Duncan’s multiple range tests.

### 4.3. Phenotypic and Paraffin Section Observations

The developmental stages of microspores were identified using the acetic acid magenta solution. Florets from D-FA99, B-FA99, D-J411 and B-J411 at the Tn were carefully removed with tweezers and observed under a Motic K400 stereomicroscope (Hong Kong, China). Mature pollen grains from anthers were stained with I_2_-KI solution to determine fertility and captured using an Olympus SZX10 (Tokyo, Japan). For paraffin section observation, anthers at the MMCs, DYAd, Td, Eun, Lun, Bn and Tn were fixed in FAA at 4 °C. After ethanol gradient dehydration and clearing in xylene, anthers of different developmental stages were embedded in paraffin. The 8 µm transverse sections were stained with 0.2% toluidine blue and images were taken by Axio Imager A2 (Zeiss, Oberkochen, Germany) [40].

### 4.4. Physiological Index Determination

In order to determine the O^2−^, H_2_O_2_, malondialdehyde (MDA) contents and peroxidase (POD) activity, anthers from different developmental stages were collected. The rate of O_2_^−^production, the H_2_O_2_ contents, MDA contents and the activities of POD were measured as described by Ba et al. and Bibi et al. [41,42].

### 4.5. Total RNA Extraction, cDNA Library Preparation, and Sequencing

The anthers collected at the Td in D-FA99, B-FA99, D-J411 and B-J411 were used to construct cDNA libraries for RNA-seq, with 12 samples (three biological replicates for each material) comprising D-FA99-TE-1, D-FA99-TE-2, D-FA99-TE-3, B-FA99-TE-1, B-FA99-TE-2, B-FA99-TE-3, D-J411-TE-1, D-J411-TE-2, D-J411-TE-3, B-J411-TE-1, B-J411-TE-2, B-J411-TE-3. Total RNA was extracted using RNAprep Pure Plant Kit (Tiangen, Beijing, China). RNA integrity and possible DNA contamination were checked by 1% agarose gel electrophoresis, RNA integrity was accurately checked with an Agilent 2100 bioanalyzer (Agilent Technologies, Santa Clara, CA, USA), and RNA purity was measured with a NanoPhotometer spectrophotometer (NanoDrop Technologies Inc., Wilmington, DE, USA). The acquisition of mRNA was mainly based on the structural characteristics of eukaryotic mRNA with a poly-(A) tail, which was enriched by Oligo (dT) magnetic beads. After the construction of the library, the Qubit2.0 fluorometer was used for preliminary quantification, the library was diluted to 1.5 ng/uL, and the insert fragment size of the library was detected by an Agilent 2100 bioanalyzer (Agilent Technologies, Santa Clara, CA, USA), and the effective concentration of the library was measured by qRT-PCR to ensure the quality of the library. According to the requirements of effective concentration and target data volume, different libraries were collected, and then Illumina sequencing (paired-end 2 × 150b) was performed. All the transcriptome raw data have been deposited in National Genomics Data Center (NGDC), Genome Sequence Archive (GSA) database with the Accession number: (CRA002624 and CRA008069). Appendix A showed the names corresponding to the samples of transcriptome raw data.

### 4.6. Quality Control and Identification of DEGs

In order to obtain high-quality clean reads, raw reads were processed as follows: (a) remove the adapter sequence in the reads; (b) remove non-AGCT-containing bases at the 5’ end before cleavage; (c) trim the ends of reads with lower sequencing quality (sequencing quality value less than Q20); (d) remove reads with a ratio of N up to 10%; (e) the small fragments with length less than 25bp after the adapter and quality trimming were discarded. The clean reads were compared with Chinese Spring wheat reference genome sequences (IWGSC_RefSeq, ver. 1.0, Shengwei Ma, Nanjing, China, https://urgi.versailles.inra.fr/download/iwgsc/IWGSC_RefSeq_Assemblies/v1.0, accessed on 10 March 2022) using HISAT2 v2.0.4 (http://ccb.jhu.edu/software/hisat2/index.shtml, accessed on 10 March 2022) [43]. The alignment of transcripts were assembled by StringTie1.3.4b (https://ccb.jhu.edu/software/stringtie/index.shtml, accessed on 10 March 2022) in a reference-based approach [44]. The fragments per kilobase million (FPKM) were used to calculate the expression levels of genes. The differential gene expression analysis using the DEseq 2 R package (1.16.1) [45], with |log_2_(foldchange)| ≥ 1 and FDR < 0.01. The *p* values obtained were adjusted by using Benjamini and Hochberg’s approach for controlling the FDR. Genes with an adjusted *p* value < 0.05 were finally assigned as DEGs.

### 4.7. Functional Annotations of DEGs and Construction of the Putative Network Model

The DEGs were functionally classified using BLAST2.2.31 software [46] based on the KOGs database (http://www.ncbi.nlm.nih.gov/KOG/, accessed on 15 March 2022). GO and KEGG functional annotations were conducted based on GO database (http://www.geneontology.org/ accessed on 15 March 2022) and the KEGG database (http://www.genome.jp/kegg/, accessed on 15 March 2022) using BLAST2.2.31. The heatmap was drawn using TBtools software [47]. We used the software of Adobe Illustrator CS5 (Adobe, CA, USA) and Science Slides Suite to draw the network model.

### 4.8. Expression Analysis of DEGs

The total RNA was isolated using the RNAprep Pure Plant Kit (Tiangen, Beijing, China) and the first-strand cDNA was obtained using a PrimeScript First-strand cDNA Synthesis Kit (Roche, Switzerland). To verify the expression levels of the DEGs, qRT-PCR with SYBR Green Dye (Gene-star Biosolutions Co., Beijing, China) was detected in QuantStudio™ 7 Flex Real Time PCR System (Applied Biosystems, USA). Sequence-specific primers (Appendix A) for DEGs and internal control *TaActin* (GenBank: GQ 339766.1) were designed using Primer-NCBI (https://www.ncbi.nlm.nih.gov/tools/primer-blast/ accessed on 20 June 2022). The qRT-PCR amplification programs were as follows: 95 °C for 10 min; followed by 45 cycles of 95 °C for 15 s, 60 °C for 30 s. The relative expression levels of DEGs were conducted using the 2^−ΔΔCT^ method and each sample was performed in three technical replicates.

## 5. Conclusions

In the present study, FA99, a novel common thermo-sensitive cytoplasmic male sterile line, was identified, which exhibited indehiscent anthers and aborted pollen grains. Through photo-thermo-sensitivity identification and paraffin section analysis, it was found that its fertility was mainly regulated by temperature, and the tetrad stage was determined as the key period of abortion. The O^2−^, H_2_O_2_, MDA contents and POD activity of FA99 in sterile conditions were significantly different from those of the control by the determination of reactive oxygen species. Further, we used the anther transcriptome data for bioinformatics analysis, and the results showed that 252 differentially expressed genes were identified, and these genes involved in “phenylalanine metabolism”, “phenylpropanoid biosynthesis”, “cutin, suberine and wax biosynthesis”, phenylalanine, tyrosine and tryptophan biosynthesis” and “citrate cycle (TCA cycle)” were closely related to pollen development. Our study provides new insight into key genes and mechanisms involved in wheat pollen development in FA99.

## Figures and Tables

**Figure 1 ijms-23-14354-f001:**
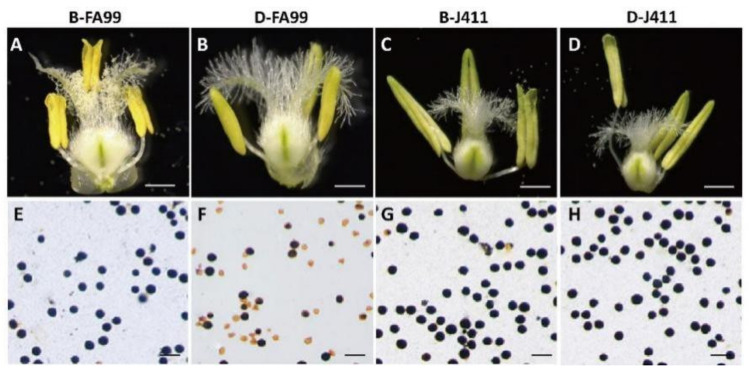
Phenotypic characteristics of FA99 and J411 in different fertility conditions. (**A**,**E**) B-FA99 (FA99 in the fertile condition); (**B**,**F**) D-FA99 (FA99 in the sterile condition); (**C**,**G**) B-J411 (J411 in the fertile condition); (**D**,**H**) D-J411 (J411 in the sterile condition); (**A**–**D**) Floret at the trinucleate stage; (**E**–**H**) I_2_-KI staining of pollen grains. Scale bars = 500 µm (**A**–**D**); 100 µm (**E**–**H**).

**Figure 2 ijms-23-14354-f002:**
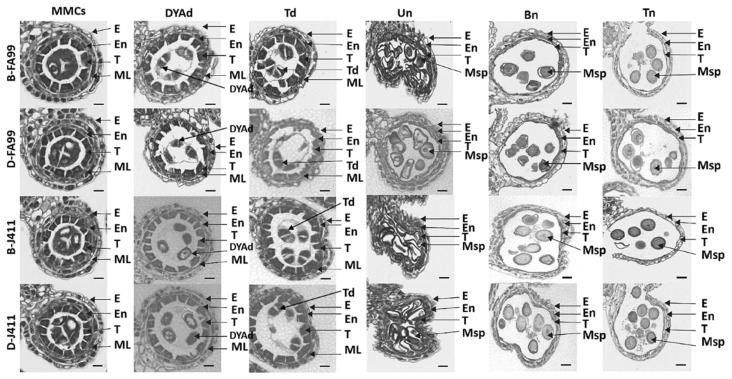
Comparative analysis of paraffin sections of FA99 and J411 anthers at different stages in different fertility environments. MMCs: microspore mother cell stage; DYAd: dyad stage; Td: tetrad stage; Un: uninucleate stage; Bn: binucleate stage; Tn: trinucleate stage; E: Epidermis; En: Endothecium; ML: Middle layer; T: Tapetum; Msp: Microspores; Scale bars = 50 µm.

**Figure 3 ijms-23-14354-f003:**
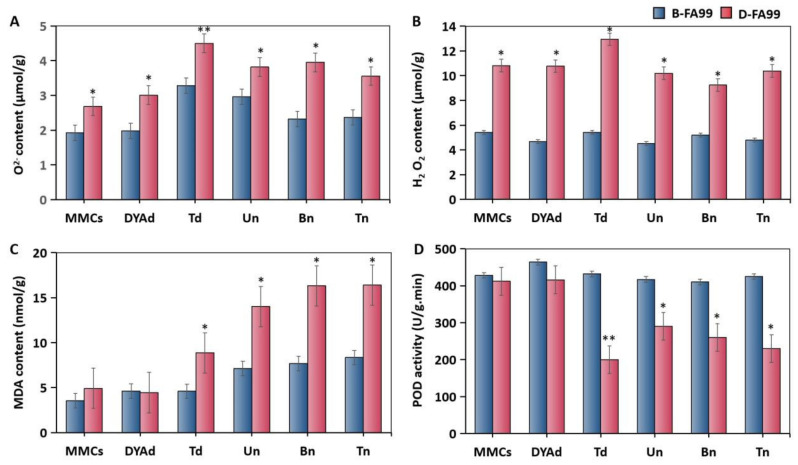
Analysis of FA99-related physiological indexes in different fertility environments. (**A**) O^2−^ contents; (**B**) H_2_O_2_ contents; (**C**) MDA contents; (**D**) POD activity. MMCs: microspore mother cell stage; DYAd: dyad stage; Td: tetrad stage; Un: uninucleate stage; Bn: binucleate stage; Tn: trinucleate stage. Students’ *t* test * *p* < 0.05, ** *p* < 0.01. Each value represents the mean ± SD (*n* = 3).

**Figure 4 ijms-23-14354-f004:**
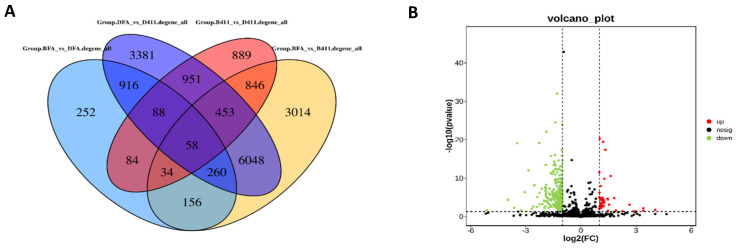
Analysis of DEGs between the samples. (**A**) Venn diagram of all DEGs; (**B**) Differences in the abundance of genes.

**Figure 5 ijms-23-14354-f005:**
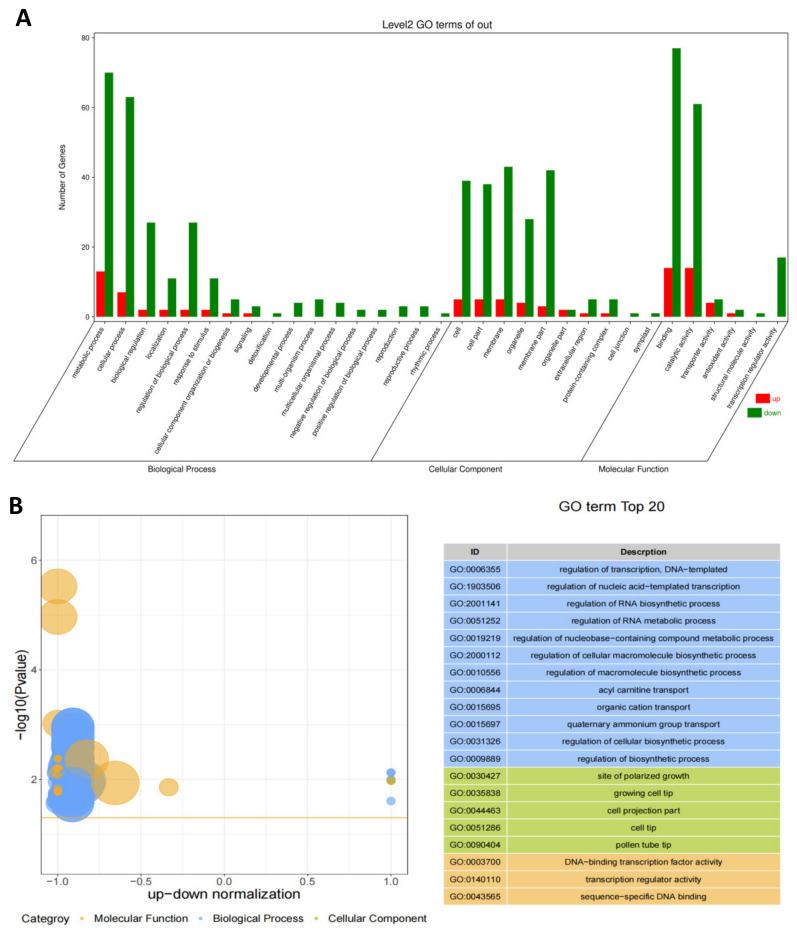
Gene Ontolgoy (GO) classifications of differentially expressed genes (DEGs) (**A**) and the Z-score bubble chart of GO enrichment analysis (**B**) in B-FA99 and D-FA99 during the tetrad stage. The ordinate is -log10 (*p* value), the abscissa is the up–down normalization value (the ratio of the difference between the number of differentially upregulated genes and the number of differentially downregulated genes to the total differential genes). The bubble size indicates the number of target genes enriched by the current GO term; the yellow line represents the threshold of *p* value = 0.05; the right side is the list of terms with the top 20 *p* values, and different colors represent different ontology.

**Figure 6 ijms-23-14354-f006:**
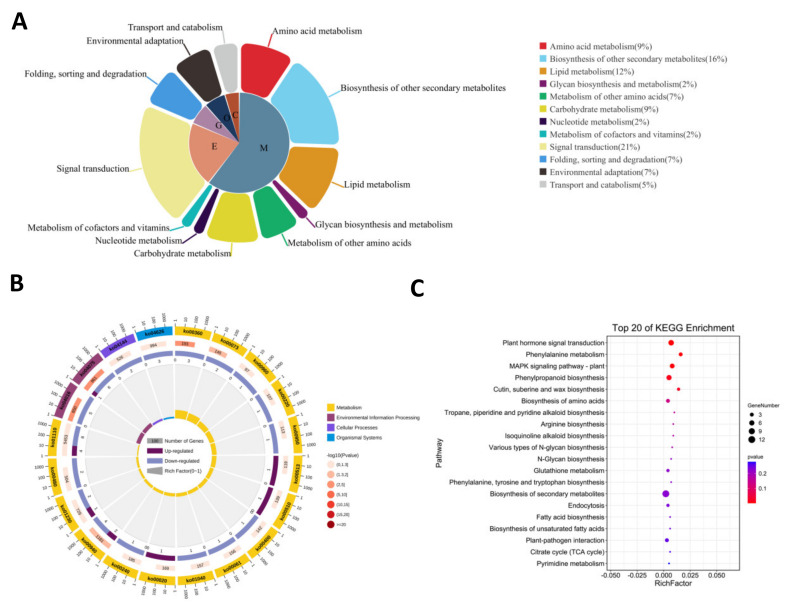
Primary and secondary classification (**A**) and the top 20 significantly enriched pathways based on KEGG (**B**,**C**). C: cellular processes; O: organismal systems; G: genetic information processing; E: environmental information processing; M: metabolism.

**Figure 7 ijms-23-14354-f007:**
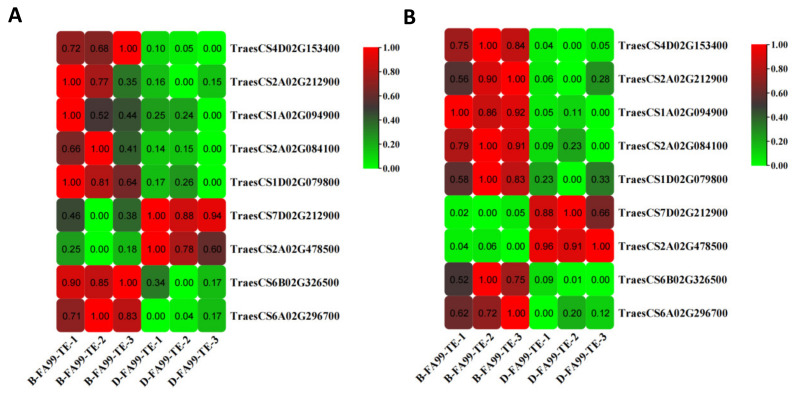
Heatmap showing the relative expression level of the nine key genes determined by RNA-seq analysis (**A**) and qRT-PCR (**B**).

**Figure 8 ijms-23-14354-f008:**
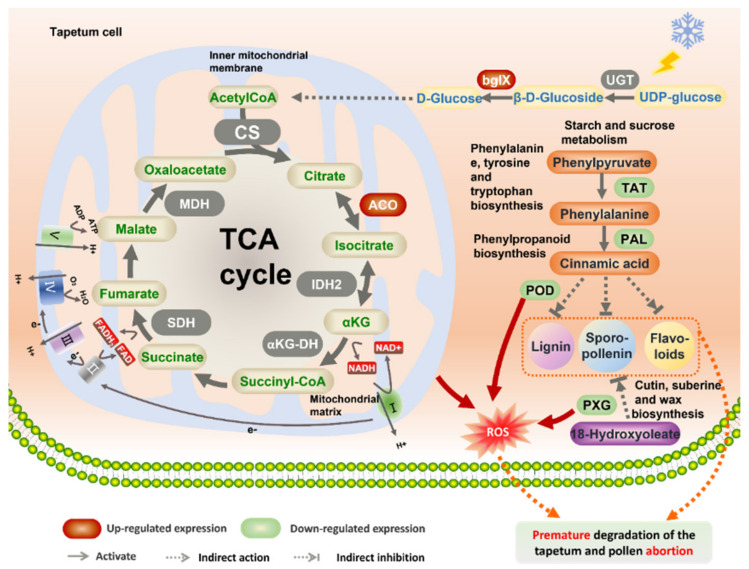
Possible transcriptome-mediated male sterility network in FA99. UGT, UDP-glucose glucosyltransferase; bglX, β-glucosidase; CS, citrate synthase; MDH, malate dehydrogenase, SDH, succinate dehydrogenase; αKG-DH, 2-oxoglutarate dehydrogenase E1 component; IDH_2_, isocitrate dehydrogenase; ACO, aconitate hydratase; FAD, flavin adenine dinucleotide; FADH_2_, flavine adenine dinucleotide, reduced; NAD, nicotinamide adenine dinucleotide; NADH, nicotinamide adenine dinucleotide; TAT, tyrosine aminotransferase; PAL, phenylalanine ammonia-lyase; POD, peroxidase; PXG, peroxygenase; ROS, reactive oxygen species; I, NADH dehydrogenase; II, succinate dehydrogenase; III, cytochrome bc1 complex; IV, cytochrome c oxidase; V, ATP synthase.

**Table 1 ijms-23-14354-t001:** Seed setting rates of FA99 and J411 in Shunyi, Beijing and Dengzhou, Henan during 2017 and 2018.

Sowing Date	Test Location 1 Shunyi, Beijing		Test Location 2 Dengzhou, Henan	
	Seeds Set in FA99 (%)	Seeds Set in J411 (%)	Seeds Set in FA99 (%)	Seeds Set in J411 (%)
30 September 2017	19.52 a	94.49 a	0.00 a	95.56 a
10 October 2017	20.66 a	97.11 a	0.42 a	98.41 a
20 October 2017	35.63 b	100.02 a	4.61 b	97.89 a
30 October 2017	—	—	6.35 b	104.54 a
10 November 2017	—	—	18.44 c	102.98 a
30 September 2018	18.29 a	95.87 a	0.00 a	97.13 a
10 October 2018	20.96 a	98.23 a	0.57 a	96.32 a
20 October 2018	37.82 b	102.10 a	3.97 b	98.47 a
30 October 2018	—	—	6.90 b	105.69 a
10 November 2018	—	—	16.85 c	103.33 a

The statistical comparison of multiple set was performed by Duncan’s multiple range tests. Different letters after the same column of data in the same year indicate significant differences between different sowing date (*p* < 0.05).

**Table 2 ijms-23-14354-t002:** Identification of photo-thermo-sensitive and fertility conversion stage.

Classification		Treatment	FA99 (%)	J411 (%)
Temperature/°C		10	0.10 a	72.24 a
		12	0.56 a	75.32 a
		14	8.51 b	78.89 a
		16	19.36 c	80.22 a
Photoperiod/(h·d^−1^)		10	0.12 a	86.13 a
		12	0.78 a	83.04 a
		14	1.01 a	80.53 a
Stage of treatment	Floret differentiation to pistil and stamen differentiation	low temperature (10 °C)	15.98 a	70.13 a
	Pistil and stamen differentiation to pollen mother cell stage		12.45 a	72.32 a
	Pollen mother cell stage to tetrad stage		0.11 b	69.87 a
	Tetrad stage to uninucleate stage		0.18 b	71.85 a

The statistical comparison of multiple set was performed by Duncan’s multiple range tests. Different letters after the same column of data indicate that there were significant differences between different temperatures, different photoperiods and different stage (*p* < 0.05).

**Table 3 ijms-23-14354-t003:** Summary and evaluation of transcriptome-sequencing data.

Samples ID	Total Reads	Clean Reads	GC Content (%)	Q30 (%)	Mapped Reads (%)
D-FA99-TE-1	117,990,620	115,339,156	53.84	92.84	86.981
D-FA99-TE-2	49,126,114	49,108,526	54.38	95.08	89.215
D-FA99-TE-3	76,933,564	74,340,224	53.76	93.39	87.315
B-FA99-TE-1	92,631,228	89,504,426	53.70	93.35	84.094
B-FA99-TE-2	84,097,450	82,116,542	53.33	93.23	89.108
B-FA99-TE-3	93,299,450	90,770,058	53.46	93.56	89.989
D-J411-TE-1	71,488,332	35,744,166	56.94	94.97	93.55
D-J411-TE-2	75,086,450	37,543,225	56.36	95.05	90.99
D-J411-TE-3	74,145,164	37,072,582	56.79	95.26	94.16
B-J411-TE-1	71,026,180	35,511,809	55.53	94.37	93.71
B-J411-TE-2	84,096,402	42,048,201	56.59	94.27	94.02
B-J411-TE-3	71,733,286	35,866,643	56.83	94.41	93.53

## Data Availability

Not applicable.

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
