# Peer review of "Comparative Transcriptome Analysis Reveals Key Insights into Fertility Conversion in the Thermo-Sensitive Cytoplasmic Male Sterile Wheat"

_ijms, 2022, doi:10.3390/ijms232214354_

Round 1
Reviewer 2 Report
Comment
I think the paper is well written, rigorous, and statistically robust.
Minor comment:
1. Section 2.4. The authors introduced two variables for the differential expression (genotype and environment): two genotypes (FA* and J*) under two different fert environments (B* and D*) and implemented mostly pairwise model framework.
It would be interesting if the authors could implement more complex model:
complex models examples:
1.genotypic DE: (B-FA*, D-FA*) vs (B-J*, D-J*)
2. residual DE: B-FA* vs B-J* vs D-FA* vs D-J*
3. 3-way DE - to determine G x E interactions
This is a suggestion and the authors are free to ignore if not part of the paper's aim.
Minor suggestion:
1. I suggest that the authors use the v2 of IWGSC. This is just a suggestion and the authors are free to ignore/consider it as I know how tedious it would be to shift to another reference genome. #line 465
2. p-value < 0.05; FDR<0.05 p values have lots of noises. why not use FDR for multiple testing Ho #line 470
3. line 477 "softwares" should be "software"
4. labels of the figures are pixilated/blurry in the manuscript (but they are clearer in the attached figures). please improve them
5. line 65-67 improve statement
6. In section 2.1, indicate statistical test used.
7. Some statements need improvement of the grammar.
Please see attached pdf file for minor edits.
